# Local $K$-means: An Efficient Optimization Algorithm And Its Generalization

## Abstract

Until now, $k$-means is still one of the most popular clustering algorithms because of its simplicity and efficiency, although it has been proposed for a long time. In this paper, we considered a variant of $k$-means that takes the $k$-nearest neighbor ($k$-NN) graph as input and proposed a novel clustering algorithm called Local K-Means (LKM). We also developed a general model that unified LKM, KSUMS, and SC, and discussed the connection among them. In addition, we proposed an efficient optimization algorithm for the unified model. Thus, not only LKM but also SC can be optimized with a linear time complexity with respect to the number of samples. Specifically, the computational overhead is $O(nk)$, where $n$ and $k$ are denote the number of samples and nearest neighbors, respectively. Extensive experiments have been conducted on 11 synthetic and 16 benchmark datasets from the literature. The effectiveness, efficiency, and robustness to outliers of the proposed method have been verified by the experimental results.

## 1 Introduction

Clustering is one of the fundamental tasks of machine learning [10]. It plays a very important role in many applications such as document analysis [6], image processing [14], and recommender system [12]. Given a dataset with $n$ samples and the number of clusters $c$, its purpose is to split these samples into $c$ disjoint groups, so that the samples within the same group are similar to each other, and the samples between different groups are not. Although there are lots of clustering algorithms have been proposed, $k$-means is still getting a lot of attention. In this paper, we proposed an efficient clustering method called local $k$-means where a $k$-NN graph is taken as input. It can be seen as a variant of traditional $k$-means. In the following, the two basic materials of our model are firstly described, and the main contributions of this article will be mentioned at the end of this section.

**Notations:** Bold capital letters and bold lowercase letters denote matrices and vectors, respectively. The symbols $n$, $d$, and $c$ are respectively used to represent the number of samples of the dataset, the number of features, and the number of clusters to construct. For matrix $\mathbf{A}$, we call it indicator matrix, if each row of it has only one element equal to 1. $\Phi^{n \times c}$ is the set of all indicator matrices.

### 1.1 $k$-means

As one of the most popular clustering algorithms, $k$-means aims to group n samples into c clusters where each sample belongs to the cluster with the nearest cluster centers. Let $\mathbf{X} = [\mathbf{x}_1, \cdots, \mathbf{x}_n]^T \in \mathbb{R}^{n \times d}$ be a collection of samples to cluster, where $\mathbf{x}_i \in \mathbb{R}^d$ denotes the $i$-th sample. Then the objective function of $k$-means can be formulated as

$$\min_{\mathcal{A}_1, \cdots, \mathcal{A}_c} \sum_{k=1}^{c} \sum_{\mathbf{x}_i \in \mathcal{A}_k} \|\mathbf{x}_i - \mathbf{m}_k\|_2^2, \tag{1}$$

Submitted to 35th Conference on Neural Information Processing Systems (NeurIPS 2021). Do not distribute.

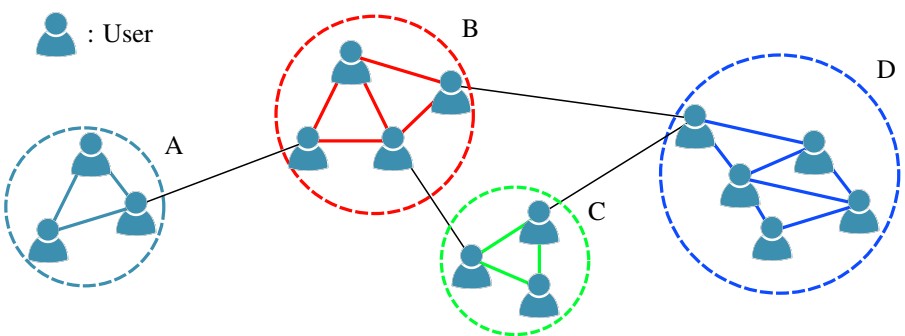

Figure 1: Community in the social network. There is a connection between two users if they know each other, in other words, the two people are friends with each other. The thicker the line, the more familiar the two users. According to the connections between users, the clustering algorithm divides them into disjoint sets. For example, a partition composed of A, B, C, and D is a satisfactory clustering result.

where $\mathcal{A}_k$ denotes the set of samples in the $i$-th cluster, $\mathcal{A}_1 \bigcup \cdots \bigcup \mathcal{A}_c = \{\mathbf{x}_i \mid i = 1, \cdots, n\}$, and $\mathbf{m}_k$ denotes the mean of samples in $\mathcal{A}_k$.

Although the problem in Eq. (1) is computationally difficult, [1] many efficient optimization algorithms where a local optimum will be found quickly have been proposed. Among them, Lloyd's algorithm is the most widely used. Let $\mathbf{Y} = [\mathbf{y}_1, \cdots, \mathbf{y}_n]^T = [\bar{\mathbf{y}}_1, \cdots, \bar{\mathbf{y}}_c] \in \mathbb{R}^{n \times c}$ be an indicator matrix, i.e.,

$$y_{ij} = \begin{cases} 1 & \mathbf{x}_i \in \mathcal{A}_j \\ 0 & \text{otherwise} \end{cases}, i = 1, \cdots, n, j = 1, \cdots, c, \tag{2}$$

the problem in Eq. (1) can be then rewritten as

$$\min_{\mathbf{Y}} \|\mathbf{X} - \mathbf{Y}\mathbf{M}\|_2^2, \tag{3}$$

where $\mathbf{M} = (\mathbf{Y}^T \mathbf{Y})^{-1} \mathbf{Y}^T \mathbf{X}$. In Lloyd's algorithm, $\mathbf{Y}$ and $\mathbf{M}$ are regarded as two independent variables and be optimized alternately.

## 1.2  Data in the form of graph

In fields such as social networks and recommendation systems, the data being studied is often presented in the form of graphs. In other words, for a single sample, we have no features to describe it, what we have is only the relationship between it and others, as shown in Figure 1.

In generally, a sparse similarity matrix $\mathbf{W} \in \mathbb{R}^{n \times n}$ can be used to describe this kind of data, i.e.,

$$w_{ij} = \begin{cases} f(\mathbf{x}_i, \mathbf{x}_j) & \text{If } \mathbf{x}_i \text{ and } \mathbf{x}_j \text{ are directly connected} \\ 0 & \text{Otherwise} \end{cases}, i, j = 1, \cdots, n, \tag{4}$$

where $f(\mathbf{x}_i, \mathbf{x}_j)$ represents the similarity between $\mathbf{x}_i$ and $\mathbf{x}_j$, and its value can be usually obtained directly.

Based on the above discussion, a $k$-means-like algorithm is proposed, which takes the $k$-NN graph as input and can be quickly optimized. In addition, we also discussed its connection with other algorithms, such as KSUMS and spectral clustering. Here, we summarize the main contributions of the article as follows

- A novel clustering algorithm called Local K-Means (LKM) is proposed. Because only the distances between the sample and its neighbors are considered, LKM is robust to outliers.
- The relationship between LKM and other algorithms (KSUMS and SC) is discussed, and a unified model is established.
- An efficient optimization algorithm for the unified model is developed, from which we find that the spectral clustering model can be optimized in the same way as LKM, which means both of them can also be optimized in $O(nk)$ time.

---

[1]Specifically, it is an np-hard problem.

## 2    Related work

A disadvantage of $k$-means is that its performance will be affected largely by the initialization of the cluster center. To this end, a lot of efforts have been made, such as [2, 4, 3]. In these methods, the cluster center is carefully initialized through a special process. In addition to the more robust clustering result, an improvement of performance can also be achieved. More related work can be found here [15, 22].

Since the computational complexity of $k$-means involves the product of the number of samples and clusters, it will be very time-consuming if the two numbers are very large. With the help of techniques that used to accelerate the nearest neighbor search, the nearest center for each sample can be quickly found without computing distances to all centers [25, 11]. [7] developed a fast implementation of $k$-means using coreset. A partition on a small coreset is computed firstly and is used as an initialization on a larger coreset. In [32], Xia et al. described each cluster by a ball and proposed Ball $k$-means which accelerated $k$-means by reducing the computation of distances between samples and centers. [13] proposed compressive $k$-means (CKM) where the centers are estimated from a sketch (a compressed representation of the original data). Once the sketch is obtained, the computational overhead is then independent of the size of the original data. Moreover, it's also a hot spot to use the advantages of GPU to shorten the time consumed by $k$-means, such as [17] and [5].

Clustering on graph data is also a hot topic. Some well-known algorithms include [19, 29, 21]. However, these algorithms often have a time complexity that increases quadratically with respect to the number of samples. To this end, many fast versions of them are proposed [33, 20, 9].

## 3    The proposed model

In our article, how to solve the problem in Eq. (1) has not been paid attention to, but some simple derivations are firstly made on it. Therefore we can analyze the meaning of the problem from the perspective of a distance graph. For convenience, we define $\mathcal{N}_k(\mathbf{x}_i) = \{\mathbf{x}_j \mid \mathbf{x}_j$ is among the $k$-nearest neighbors of $\mathbf{x}_i$ or $\mathbf{x}_i$ is among the $k$-nearest neighbors of $\mathbf{x}_j\}$, and start from the following equivalent form of $k$-means

$$\min_{\mathcal{A}_1,\cdots,\mathcal{A}_c} \sum_{k=1}^{c} \frac{1}{|\mathcal{A}_k|} \sum_{\mathbf{x}_i,\mathbf{x}_j \in \mathcal{A}_k} \|\mathbf{x}_i - \mathbf{x}_j\|_2^2, \tag{5}$$

With the help of the definition of $\mathbf{Y}$ in Eq. (2), problem (5) can be equivalently expressed as follows

$$\min_{\mathbf{Y} \in \Phi^{n \times c}} diag\left((\mathbf{Y}^T\mathbf{Y})^{-1}\right)^T diag\left(\mathbf{Y}^T\mathbf{D}\mathbf{Y}\right), \tag{6}$$

$$\Leftrightarrow \min_{\mathbf{Y} \in \Phi^{n \times c}} Tr\left((\mathbf{Y}^T\mathbf{Y})^{-1}\mathbf{Y}^T\mathbf{D}\mathbf{Y}\right), \tag{7}$$

where $diag(\mathbf{A}) = [a_{11}, \cdots, a_{nn}]^T$. Obviously, if we only consider the distances between the sample and its neighbors, then the problem in Eq. (7) can be expressed as

$$\min_{\mathbf{Y} \in \Phi^{n \times c}} Tr\left((\mathbf{Y}^T\mathbf{Y})^{-1}\mathbf{Y}^T\mathbf{D}^{(k)}\mathbf{Y}\right), \tag{8}$$

with

$$\mathbf{d}_{ij}^{(k)} = \begin{cases} \|\mathbf{x}_i - \mathbf{x}_j\|_2^2 & \text{if } \mathbf{x}_i \in \mathcal{N}_k(\mathbf{x}_j) \\ \gamma & \text{Otherwise} \end{cases}, \tag{9}$$

where $\gamma$ is the maximum value of set $\{\|\mathbf{x}_i - \mathbf{x}_j\|_2^2 \mid \mathbf{x}_i \in \mathcal{N}_k(\mathbf{x}_j), i = 1, \cdots, n\}$. The Equation (8) is the final objective function of LKM.

From the discussion in Section 1.2, we know that only the similarity instead of the distance between samples can be obtained directly in graph data. Fortunately, in practical applications, we can convert the similarity to dissimilarity by

$$r_{ij} = \begin{cases} -log(s_{ij}) & 0 < s_{ij} \\ \beta & s_{ij} = 0 \end{cases}, \tag{10}$$

where $s_{ij}$ is the normalized[2] similarity between $\mathbf{x}_i$ and $\mathbf{x}_j$, $\beta$ is the maximum value of set $\{-log(s_{ij}) \mid i, j = 1, \cdots, n\}$. Then the dissimilarity can be used to replace the distance in the model.

---

[2] $s_{ij} \in [0, 1]$

## 3.1 Generalization

It is not difficult to find that LKM, KSUMS [23], and Ratio-cut [29] can all be represented uniformly by the following model

$$\min_{\mathbf{Y} \in \Phi^{n \times c}} Tr\left((\mathbf{Y}^T \mathbf{Y})^{-p} \mathbf{Y}^T \mathbf{G}^{(k)} \mathbf{Y}\right),$$ (11)

where $g_{ij}^{(k)}$ denotes the dissimilarity or distance between $\mathbf{x}_i$ and $\mathbf{x}_j$, and $p >= 0$ is a parameter. The meaning of $p$ will be explored in future work.

**Instances of KSUMS and LKM**: The objective function of KSUMS is

$$\min_{\mathbf{Y} \in \Phi^{n \times c}} Tr\left(\mathbf{Y}^T \mathbf{D}^{(k)} \mathbf{Y}\right),$$ (12)

where $\mathbf{D}^{(k)}$ takes the same expression as that in LKM. Let $g_{ij}^{(k)}$ be setted by Eq. (9), the problem (11) is identical with KSUMS (12) if $p = 0$, and is identical with LKM if $p = 1$.

**Instance of Ratio-cut:** Benefiting from the introduction of $\mathbf{Y}$, the problem of ratio-cut (an algorithm that belongs to the spectral clustering (SC) family) can be expressed as

$$\min_{\mathbf{Y} \in \Phi^{n \times c}} Tr\left((\mathbf{Y}^T \mathbf{Y})^{-1} \mathbf{Y}^T (\mathbf{\Delta} - \mathbf{W}) \mathbf{Y}\right),$$ (13)

where $\mathbf{\Delta}$ is a diagonal matrix, $\Delta_{ii} = \sum_{j=1}^{n} w_{ij}$. In generally, the similarity matrix $\mathbf{W}$ can be determined by heat kernel, i.e., $w_{ij} = e^{-\frac{\|\mathbf{x}_i - \mathbf{x}_j\|_2^2}{t}}$ if $\mathbf{x}_i \in \mathcal{N}_k(\mathbf{x}_j)$, $w_{ij} = 0$ otherwise. Therefore the problem (11) is equivalent with ratio-cut, if $p = 1$ and $g_{ij}^{(k)}$ is setted by

$$\mathbf{g}_{ij}^{(k)} = \begin{cases} \sum_{j=1}^{n} w_{ij} & i = j \\ -w_{ij} & i \neq j, \text{ and } \mathbf{x}_i \in \mathcal{N}_k(\mathbf{x}_j) \\ 0 & \text{Otherwise} \end{cases}.$$ (14)

## 3.2 Optimization

From the discussion above, we know that the problem of LKM can be expressed by Eq. (11) with $p = 1$. Therefore, an optimization algorithm for problem (11) instead of problem (8) is developed. To begin with, some notations are presented as follows

$$s_i \triangleq \bar{\mathbf{y}}_i^T \mathbf{G}^{(k)} \bar{\mathbf{y}}_i, \quad i = 1, \cdots, c,$$ (15)

$$n_i \triangleq \bar{\mathbf{y}}_i^T \bar{\mathbf{y}}_i, \quad i = 1, \cdots, c,$$ (16)

the problem (11) then becomes

$$\min_{\mathbf{Y} \in \Phi^{n \times c}} Obj(\mathbf{Y}), \text{ with } Obj(\mathbf{Y}) = \sum_{i=1}^{c} \frac{s_i}{n_i^p}.$$ (17)

In the following derivation, the $i$-th row of $\mathbf{Y}$ (i.e., $\mathbf{y}_i$) is regarded as the variable to be optimized while others are fixed, and $\mathbf{y}_i = \mathbf{e}_\alpha$ before updated. Thus $\mathbf{y}_i$ can be updated by

$$\mathbf{y}_i = \mathbf{e}_\beta, \quad \beta = \arg\min_{j} Obj(\mathbf{y}_i = \mathbf{e}_j) - Obj(\mathbf{y}_i = \mathbf{0}),$$ (18)

where $\mathbf{e}_i = [0, \cdots, 1, \cdots, 0]$ be a vector with all elements equal to 0, except the $i$-th, which is 1, and $\mathbf{0}$ is the column vector of all zeros,

Because $Obj(\mathbf{y}_i = \mathbf{0})$ is constant, the above formula holds. According to Eq. (17), we have

$$Obj(\mathbf{y}_i = \mathbf{e}_j) - Obj(\mathbf{y}_i = \mathbf{0}) = \begin{cases} \frac{s_j + b_j}{(n_j+1)^p} - \frac{s_j}{n_j^p} & j \neq \alpha \\ \frac{s_j}{n_j^p} - \frac{s_j - b_j}{(n_j-1)^p} & j = \alpha \end{cases}, j = 1, \cdots, c,$$ (19)

with

$$b_j = \begin{cases} 2\sum_{\mathbf{x}_l \in \mathcal{A}_j} g_{il}^{(k)} + g_{ii}^{(k)} & j \neq \alpha \\ 2\sum_{\mathbf{x}_l \in \mathcal{A}_j} g_{il}^{(k)} - g_{ii}^{(k)} & j = \alpha \end{cases},$$ (20)

---

**Algorithm 1:** An efficient program for solving problem (11).

---

Note: The vector $\mathbf{y} \in \mathbb{R}^n$ denotes the clustering result, i.e., $y_i$ is the cluster that $\mathbf{x}_i$ belongs to. The Eq. (15), (16), and (20) involved in the algorithm have high computational complexity, but these can be computed more efficiently if the sparsity of $\mathbf{G}^{(k)}$ is considered. See the supplementary material for a more detailed algorithm;

**Data:** Sparse matrix $^3\mathbf{G}^{(k)} \in \mathbb{R}^{n \times n}$, the number of cluster $c$

**Result:** The clustering result $\mathbf{y}$

Initialize $\mathbf{y}$ randomly;

Compute vector $\mathbf{s}$ and $\mathbf{n}$ by Eq. (15) and (16), respectively;

**while** *not converge* **do**

    **for** $i = 1, \cdots, n$ **do**

        Compute $b_j$ by Eq. (20) for $j \in \mathcal{B}_i$;

        Compute $Obj(y_i = j) - Obj(y_i = 0)$ by Eq. (19) for $j \in \mathcal{B}_i$;

        Update $y_i$ by Eq. (18);

        Update $\mathbf{s}$ and $\mathbf{n}$ by Eq. (21) and Eq. (22), respectively;

---

Benefiting from the sparsity of $\mathbf{G}^{(k)}$, it takes $O(nk)$, $O(k+c)$, and $O(k)$ time to compute $\mathbf{s}$, $\mathbf{b}$, and $\mathbf{n}$, respectively. Therefore, the proposed optimization algorithm has a computational complexity of $O(n^2k + nc)$, which is unbearable, for large-scale datasets. However, if the variables $\mathbf{s}$ and $\mathbf{n}$ are computed in advance and updated following the update of $y_i$, then the computational complexity of the algorithm can greatly be reduced. The update rules for $\mathbf{s}$ and $\mathbf{n}$ are as follows

$$s_\alpha \Leftarrow s_\alpha - b_\alpha, \quad s_\beta \Leftarrow s_\beta + b_\beta, \tag{21}$$

$$n_\alpha \Leftarrow n_\alpha - 1, \quad n_\beta \Leftarrow n_\beta + 1, \tag{22}$$

Thus, the computational complexity of the optimization algorithm is $O(n(k+c))$.

**On more step** From Eq. (11), we know that only the information of pair $(\mathbf{x}_i, \mathbf{x}_j)$ is considered in the model, and there are at most $2nk$ such pairs. For convenience, we assume that there are exactly $2k$ such pairs for each sample $\mathbf{x}_i$, i.e., $2k = |\{(\mathbf{x}_i, \mathbf{x}_j) \mid \mathbf{x}_j \in \mathcal{N}_k(\mathbf{x}_i) \text{ or } \mathbf{x}_i \in \mathcal{N}_k(\mathbf{x}_j)\}|$. For cluster $j$, we call it an element of $\mathcal{B}_i$ ($j \in \mathcal{B}_i$), if there is at least one sample in cluster $j$ belongs to $\mathcal{N}_k(\mathbf{x}_i)$ or $\mathbf{x}_i$ belongs to the set of neighbors of these samples. Based on the assumption and notations above, we know that when updating $\mathbf{y}_i$ by Eq. (18), the size of $\mathcal{B}_i$ is at most $2k$. However, it does not make sense to group the sample $\mathbf{x}_i$ into cluster $j \notin \mathcal{B}_i$, from the perspective of the performance. Therefore, we only need to pay attention to the cases where $j \in \mathcal{B}_i$. Thus, the computational complexity of the optimization algorithm can be reduced to $O(nk)$.

**Time and space complexity** From Algorithm 1, we can see that the memory is mainly occupied by the matrix $\mathbf{G}^{(k)} \in \mathbb{R}^{n \times n}$, which is equivalent to a sparse matrix, and contains at most $2nk$ non-constants. The memory overhead caused by other variables is $O(n)$ at most. For example, $\mathbf{y}$, $\mathcal{B}_i$, and $\mathbf{s}$ require $O(n)$, $O(k)$, and $O(c)$ memory, respectively. Thus the memory overhead of LKM is $O(nk)$. Benefiting from the sparsity of $\mathbf{G}^{(k)}$, Eq. (15), (16), and (20) can all be calculated more efficiently. Specifically, only $O(nk)$, $O(n)$, and $O(k)$ time are needed respectively, please refer to the supplementary materials for details. After $y_i$ is updated, only $O(1)$ time is needed to update variables $\mathbf{s}$ and $\mathbf{n}$. Thus, the computational complexity of LKM is $O(nk)$.

## 4   Experiments

In this section, the performance of the proposed algorithm, LKM, is verified on eleven synthetic datasets and sixteen benchmark datasets. The rest of this section is organized as follows: First, experiments on synthetic datasets are shown. In short, Mickey, Outlier, and family of Grid datasets are used to verify the effectiveness, robustness, and efficiency of LKM, respectively. Then, we compare 7 popular clustering algorithms with LKM on 16 benchmark datasets, to evaluate the performance of the proposed algorithm.

---

$^3$Strictly speaking, $\mathbf{G}^{(k)}$ is not a sparse matrix. However, at most $2nk$ values in $\mathbf{G}^{(k)}$ are not equal to $\lambda$, so it can be regarded as a sparse matrix.

## 4.1 Experiments conducted on synthetic datasets

**Experiment on "Mickey"**   To verify the effectiveness of LKM, a synthetic dataset called "Mickey" is constructed. The distribution of points is shown in Figure 2(a). The triangles representing the means of the clusters are not points of the datasets.

From Figure 2(b) and 2(c), we found that The proposed method LKM successfully found the cluster structure, but $k$-means did not. $k$-means still cannot find the correct structure, even with the initialization of the ground truth label. Because the distance between point 1 and the blue triangle (mean of all blue points), $d_1$ is greater than the distance between point 1 and the orange triangle (mean of all orange points), $d_2$, $k$-means will group it into the blue cluster instead of orange. Therefore, $k$-means cannot handle datasets like this.

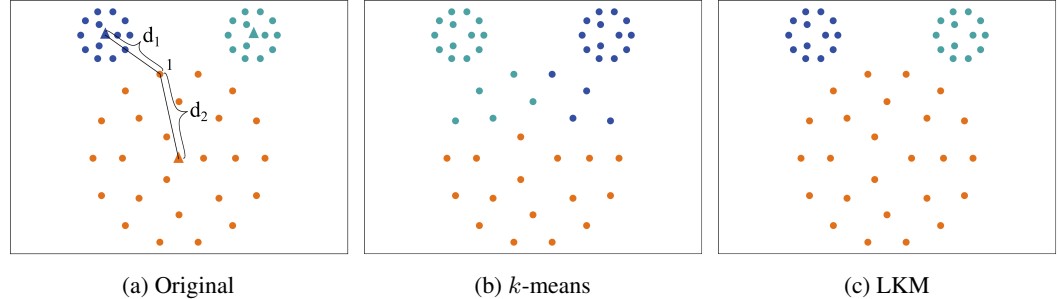

(a) Original                      (b) $k$-means                      (c) LKM

Figure 2: The performance of $k$-means and LKM on "Mickey".

**Experiment on "Outlier"**   In order to verify the robustness of our method, we construct a dataset called "Outlier". It consists of four clusters with centers $(0, 0)$, $(0, 5)$, $(5, 0)$, and $(5, 5)$, and an outlier with the coordinate of $(100, 100)$. The distance between outlier $A$ and other points is not as close as shown in Figure 3. From Figure 3(b) and 3(c), we can see that the performance of $k$-means is severely affected by the outlier $A$, while the performance of LKM is not. In $k$-means, the center of the cluster containing abnormal points will largely shift towards the direction of the abnormal points, resulting in poor performance. In LKM, the distance between $\mathbf{x}_i$ and $\mathbf{x}_j$ is not calculated if $\mathbf{x}_j \notin \mathcal{N}_k(\mathbf{x}_i)$, but a parameter $\lambda$ is used instead, so ideally, the distance between any two points belonging to different clusters is $\lambda$. In other words, for the sample point $\mathbf{x}_i$, there is no difference between the outlier and the samples that do not belong to $\mathcal{N}_k(\mathbf{x}_i)$.

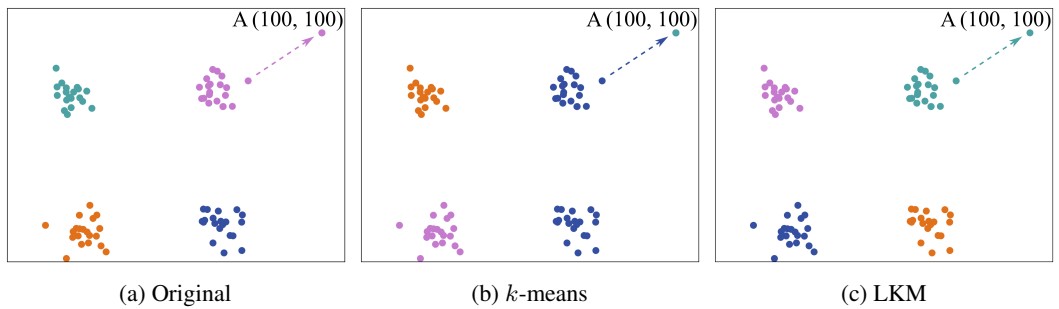

(a) Original                      (b) $k$-means                      (c) LKM

Figure 3: The performance of $k$-means and LKM on "Outlier".

**Experiments on the family of "Grid"**   In order to verify the efficiency of LKM, in this paragraph, 9 synthetic datasets called Toy-1, Toy-2, $\cdots$, Toy-9 are constructed. These datasets share the same structure, and their distributions are similar to that shown in Figure 4. In these datasets, each cluster is always composed of 10 points generated by Gaussian distribution. Since the time complexity of LKM and $k$-means is closely related to the number of points, we set different sizes for these data sets, ranging from 1960 to 125440. The number of clusters and the standard deviation involved in the Gaussian distribution for each dataset is shown in Table 1.

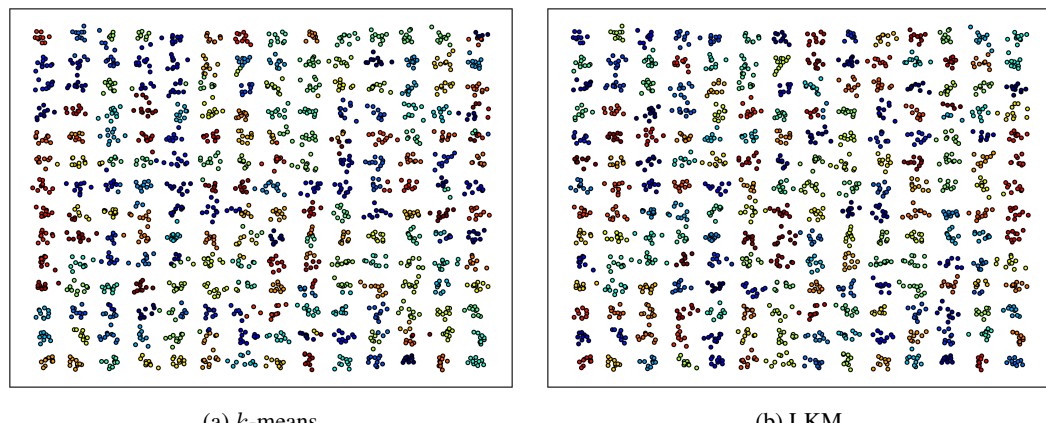

(a) $k$-means                    (b) LKM

Figure 4: The performance of $k$-means and LKM on Toy-1.

Table 1: Performance of $k$-means and LKM

| Datasets | # Clusters | $3\sigma$ | Precision | | Recall | | $F_1$ score | |
|---|---|---|---|---|---|---|---|---|
| | | | $k$-means | LKM | $k$-means | LKM | $k$-means | LKM |
| Toy-1 | 196 | 0.5 | 0.854 | **0.975** | 0.915 | **0.983** | 0.883 | **0.979** |
| Toy-2 | 196 | 0.6 | 0.834 | **0.948** | 0.885 | **0.957** | 0.859 | **0.953** |
| Toy-3 | 196 | 0.7 | 0.785 | **0.874** | 0.828 | **0.889** | 0.806 | **0.881** |
| Toy-4 | 3136 | 0.5 | 0.856 | **0.981** | 0.918 | **0.988** | 0.886 | **0.984** |
| Toy-5 | 3136 | 0.6 | 0.832 | **0.947** | 0.881 | **0.957** | 0.856 | **0.952** |
| Toy-6 | 3136 | 0.7 | 0.783 | **0.883** | 0.825 | **0.893** | 0.803 | **0.888** |
| Toy-7 | 12544 | 0.5 | 0.855 | **0.982** | 0.917 | **0.988** | 0.885 | **0.985** |
| Toy-8 | 12544 | 0.6 | 0.833 | **0.948** | 0.882 | **0.957** | 0.857 | **0.952** |
| Toy-9 | 12544 | 0.7 | 0.785 | **0.884** | 0.826 | **0.896** | 0.805 | **0.890** |

Table 2: Time (s) consumed by $k$-means and LKM

| Datasets | FLK | | | | $k$-means | | Speed-up |
|---|---|---|---|---|---|---|---|
| | Ball-Tree | Algo. 1 | # Iter. | Total | # Iter. | Total | |
| Toy-1 | 6.26E-03 | 1.30E-03 | 3.96 | 7.56E-03 | 13.12 | 5.97E-03 | 1.39E+00 |
| Toy-2 | 6.54E-03 | 1.66E-03 | 5.66 | 8.20E-03 | 14.32 | 5.57E-03 | 1.33E+00 |
| Toy-3 | 6.27E-03 | 1.73E-03 | 5.96 | 8.00E-03 | 15.32 | 6.00E-03 | 1.35E+00 |
| Toy-4 | 1.34E-01 | 2.64E-02 | 5.80 | 1.60E-01 | 14.68 | 2.00E+00 | 3.00E+01 |
| Toy-5 | 1.37E-01 | 3.32E-02 | 7.64 | 1.70E-01 | 16.62 | 2.27E+00 | 3.15E+01 |
| Toy-6 | 1.39E-01 | 3.98E-02 | 9.40 | 1.79E-01 | 18.50 | 2.55E+00 | 3.25E+01 |
| Toy-7 | 6.50E-01 | 1.35E-01 | 7.20 | 7.85E-01 | 16.22 | 3.89E+01 | 1.28E+02 |
| Toy-8 | 6.04E-01 | 1.64E-01 | 9.08 | 7.68E-01 | 17.58 | 4.21E+01 | 1.33E+02 |
| Toy-9 | 6.18E-01 | 1.95E-01 | 10.96 | 8.13E-01 | 18.88 | 4.50E+01 | 1.34E+02 |

In Table 2, the column named "Ball-Tree" represents the time it takes to construct the graph required by LKM through Ball-tree with $k = 20$. The column named "# Iter" denotes the number of iterations required for the algorithm to converge. The total time of LKM refers to the sum of the time consumed by Ball-Tree and Algorithm 1. The speed-up is the ratio of the time consumed by each iteration of $k$-means to the time consumed by each iteration of Algorithm 1. Both $k$-means and LKM were run 50 times, and the average results were reported.

As shown in Table 2, Algorithm 1 consumes a significantly shorter time than $k$-means, which is more obvious on datasets with more clusters. The main reason is that when $y_i$ is going to update, only the case where $j \in \mathcal{B}_i$ is considered. In addition, LKM has a significant improvement in terms of the quality of the clustering result, compared to $k$-means, as shown in Table 1 and Figure 4.

## 4.2 Experiments conducted on benchmark datasets

### 4.2.1 Datasets

Sixteen benchmark datasets are used including LFW [8], CPLFW [34], CALFW [35], FERET [24], Colon [1], MUCT [18], CMUPIE [30], CFPW [27], Dexter, Madelon, GTDB, FaceV5, Mpeg7, Olivetti, Yale, and Umist. All facial datasets are processed by the way [23]. For those non-facial datasets, PCA [31] is adopted and some components are selected such that the amount of variance is greater than 95% if the dimensionality of the datasets is larger than 1024. The names of datasets are all linked to where the dataset can be download. The introduction to these datasets can be found in the supplemental material.

### 4.2.2 Baselines and experimental settings

We compare LKM with several clustering algorithms, including AGCI [33], FINCH [26], $k$-means [16], KSUMS [23], RCC [28], SC [29], and FCDMF [20]. For graph-based methods, i.e., KSUMS, RCC, and SC, the number of nearest neighbors, $k$, is fixed at 20. For anchor-based methods, AGCI and FCDMF, the number of anchors is always set by $m = min(n/2, 1024)$. Whether $k$-NN graph or anchor graph, heat-kernel is always adopted to construct the graph. In FINCH, we take the clustering result with the number of clusters closest to the number of ground truth clusters as the final clustering result. In RCC, the threshold to assign points together in a cluster is tuned from $\{0.1, 0.3, 0.5, 0.7, 0.9\}$. $K$-means is initialized in a random way and the step of $k$-means involved in AGCI and SC share the same configuration with $k$-means itself. If the performance of the algorithm is related to the initialization, we run it repeatedly 50 times and report the average performance.

We run all methods on an Arch machine with i7-8700 CPU (3.20 GHz), 32 GB main memory.

### 4.2.3 Experimental results

Clustering ACCuracy (ACC), Normalized Mutual Information (NMI), and Adjusted Rand index (ARI) are used to evaluate the performance of these algorithms. From Table 3, we can clearly see that: (1) In most cases LKM has achieved the highest performance comparing to several state-of-the-art algorithms, which verified the effectiveness of the proposed algorithm. Specifically, LKM exceeds the second-best results 24.4%, 4.6%, 4.8%, 1.5% and 1.3% on CALFW, LFW, Umist, Olivetti, and CMU respectively, in terms of ACC. Under the metrics of NMI and ARI, we can come to similar results. (2) Although only slight improvements LKM has achieved over many datasets compared to the second-best results, the computational complexity of LKM is much lower than that of most algorithms, which is an important property of LKM. (3) RCC has poor performance on FaceV5, CMU, GTdb, Umist, and Yale, which may be caused largely by an inappropriate threshold, while only one parameter (the number of neighbors) is needed in LKM, is an integer and easy to tune. In addition, the influence of parameter $k$ (the number of neighbors) on clustering performance has been studied, and the results are shown in the supplemental material.

## 5 Conclusions

In this paper, we devote ourselves to an unsupervised learning problem, clustering. An efficient clustering algorithm called Local K-Means (LKM) was proposed. It can be seen as a variant of $k$-means that takes the $k$-NN graph as input. We also discussed a general model that unified LKM, KSUMS, and SC. Thus the connection among them can be easily established. In addition, we developed an efficient optimization algorithm for the unified model, so that not only LKM but also SC can be optimized in $O(nk)$ time, which is very important for large-scale datasets, especially for these datasets with a large number of clusters. In order to verify the advantages of LKM, extensive experiments on eleven synthetic and sixteen benchmark datasets are conducted, and the results have shown the effectiveness, efficiency, and robustness of our model.

**Limitations** In some cases where $k$-NN graphs are not available, our algorithm cannot work, in other words, a graph construction algorithm is necessary. Although many methods have been proposed, it is still very difficult to effectively construct an approximate $k$-NN graph if the number of features is large. Thus, in these situations, the graph construction algorithm will produce a $k$-NN graph of poor quality that would lead to poor performance of clustering results.

Table 3: Performance on benchmark datasets

| Datasets | Met. | AGCI | FCDMF | FIN | $k$-means | KSUMS | RCC | SC | LKM |
|---|---|---|---|---|---|---|---|---|---|
| LFW | ACC | 0.460 | 0.450 | 0.373 | 0.460 | 0.454 | 0.551 | 0.424 | **0.597** |
| | NMI | 0.866 | 0.860 | 0.711 | 0.866 | 0.850 | 0.805 | 0.703 | **0.893** |
| | ARI | 0.063 | 0.078 | 0.008 | 0.063 | 0.037 | **0.592** | 0.010 | 0.100 |
| CALFW | ACC | 0.599 | 0.399 | 0.504 | 0.599 | 0.419 | 0.573 | 0.560 | **0.843** |
| | NMI | 0.887 | 0.859 | 0.696 | 0.888 | 0.878 | 0.886 | 0.754 | **0.971** |
| | ARI | 0.187 | 0.084 | 0.007 | 0.190 | 0.098 | 0.373 | 0.005 | **0.729** |
| CPLFW | ACC | 0.537 | 0.355 | 0.584 | 0.546 | 0.738 | **0.745** | 0.527 | 0.742 |
| | NMI | 0.770 | 0.689 | 0.613 | 0.772 | **0.889** | 0.857 | 0.733 | 0.865 |
| | ARI | 0.209 | 0.167 | 0.012 | 0.208 | **0.627** | 0.201 | 0.089 | 0.333 |
| FaceV5 | ACC | 0.730 | 0.517 | 0.535 | 0.731 | 0.934 | 0.069 | 0.621 | **0.938** |
| | NMI | 0.930 | 0.829 | 0.829 | 0.931 | 0.979 | 0.105 | 0.812 | **0.983** |
| | ARI | 0.605 | 0.280 | 0.290 | 0.621 | 0.899 | 0.001 | 0.070 | **0.910** |
| CFPW | ACC | 0.537 | 0.355 | 0.584 | 0.546 | 0.738 | **0.745** | 0.527 | 0.742 |
| | NMI | 0.770 | 0.689 | 0.613 | 0.772 | **0.889** | 0.858 | 0.733 | 0.865 |
| | ARI | 0.209 | 0.167 | 0.012 | 0.208 | **0.627** | 0.202 | 0.089 | 0.333 |
| CMU | ACC | 0.185 | 0.154 | 0.165 | 0.182 | 0.286 | 0.015 | 0.285 | **0.299** |
| | NMI | 0.409 | 0.372 | 0.306 | 0.407 | 0.571 | 0.000 | 0.552 | **0.582** |
| | ARI | 0.079 | 0.063 | 0.018 | 0.077 | 0.192 | 0.000 | 0.173 | **0.201** |
| Colon | ACC | 0.690 | 0.581 | 0.629 | 0.608 | 0.635 | 0.581 | 0.737 | **0.748** |
| | NMI | 0.178 | 0.010 | 0.129 | 0.094 | 0.108 | 0.045 | 0.143 | **0.259** |
| | ARI | 0.208 | 0.011 | 0.249 | 0.078 | 0.110 | -0.05 | 0.210 | **0.317** |
| Dexter | ACC | 0.579 | **0.627** | 0.153 | 0.596 | 0.584 | 0.490 | 0.567 | 0.612 |
| | NMI | 0.077 | **0.124** | 0.080 | 0.091 | 0.024 | 0.051 | 0.015 | 0.123 |
| | ARI | 0.035 | **0.063** | 0.011 | 0.042 | 0.031 | 0.002 | 0.017 | 0.050 |
| FERET | ACC | 0.522 | 0.378 | 0.495 | 0.521 | 0.546 | **0.661** | 0.463 | 0.621 |
| | NMI | 0.822 | 0.734 | 0.686 | 0.822 | 0.839 | 0.714 | 0.735 | **0.863** |
| | ARI | 0.354 | 0.211 | 0.039 | 0.353 | 0.439 | 0.022 | 0.036 | **0.520** |
| GTdb | ACC | 0.454 | 0.419 | 0.391 | 0.459 | 0.533 | 0.047 | 0.491 | **0.541** |
| | NMI | 0.658 | 0.634 | 0.579 | 0.661 | 0.690 | 0.032 | 0.666 | **0.697** |
| | ARI | 0.313 | 0.282 | 0.211 | 0.319 | 0.382 | 0.002 | 0.314 | **0.387** |
| Madelon | ACC | 0.517 | 0.513 | 0.456 | 0.521 | 0.529 | 0.500 | 0.507 | **0.534** |
| | NMI | 0.003 | 0.001 | 0.001 | **0.005** | **0.005** | 0.000 | 0.000 | **0.005** |
| | ARI | 0.004 | 0.000 | 0.000 | **0.006** | **0.006** | 0.000 | 0.000 | **0.006** |
| Mpeg7 | ACC | 0.463 | 0.445 | 0.442 | 0.462 | 0.539 | 0.429 | 0.462 | **0.552** |
| | NMI | 0.660 | 0.650 | 0.617 | 0.666 | 0.720 | 0.701 | 0.657 | **0.721** |
| | ARI | 0.278 | 0.295 | 0.153 | 0.291 | **0.414** | 0.452 | 0.220 | 0.346 |
| MUCT | ACC | 0.732 | 0.741 | 0.972 | 0.722 | **0.982** | 0.754 | 0.627 | 0.979 |
| | NMI | 0.928 | 0.922 | 0.991 | 0.923 | 0.992 | 0.922 | 0.791 | **0.995** |
| | ARI | 0.612 | 0.698 | 0.971 | 0.586 | 0.976 | 0.700 | 0.093 | **0.980** |
| Olivetti | ACC | 0.509 | 0.407 | 0.480 | 0.510 | 0.569 | 0.550 | 0.527 | **0.584** |
| | NMI | 0.722 | 0.643 | 0.674 | 0.718 | 0.758 | **0.780** | 0.723 | 0.768 |
| | ARI | 0.366 | 0.263 | 0.323 | 0.366 | 0.443 | 0.387 | 0.364 | **0.456** |
| Umist | ACC | 0.413 | 0.412 | 0.468 | 0.416 | 0.450 | 0.083 | 0.431 | **0.516** |
| | NMI | 0.626 | 0.589 | 0.673 | 0.628 | 0.641 | 0.000 | 0.634 | **0.690** |
| | ARI | 0.320 | 0.300 | 0.375 | 0.317 | 0.355 | 0.000 | 0.323 | **0.428** |
| Yale | ACC | 0.395 | 0.344 | 0.339 | 0.397 | 0.443 | 0.067 | 0.405 | **0.452** |
| | NMI | 0.448 | 0.398 | 0.358 | 0.455 | 0.495 | 0.000 | 0.456 | **0.498** |
| | ARI | 0.187 | 0.139 | 0.119 | 0.196 | 0.234 | 0.000 | 0.194 | **0.239** |

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
