# OpenReview forum: "Local $K$-means: An Efficient Optimization Algorithm And Its Generalization"
_NeurIPS.cc/2021/Conference — NeurIPS 2021 Submitted_

### Official Review · Reviewer_bmZT · 2021-06-25

**Rating:** 4
**Confidence:** 4

**Summary:**

The authors use a reformulation of the k-means objective to propose an algorithm that minimizes that objective (or some relaxation thereof), and then show some experiments that indicate promising clustering performance.  Currently I believe the paper needs revision before acceptance: the empirical results are not particularly compelling as they omit runtimes on real-world datasets, and the theoretical results do not factor in or bound the number of iterations of the algorithm.

**Ethical Concerns:**

I have no ethical concerns.

**Limitations And Societal Impact:**

I don't think that a new clustering algorithm has either positive or negative societal impact.  So everything seems fine here.

**Main Review:**

Hello authors,

Thank you for your submission.  I enjoyed reading it and I think there are some nice ideas contained in the paper.  However, I have a number of concerns that keep me from recommending acceptance for this paper.  Primarily, my biggest concern is the given runtime bound (which I find inaccurate as it does not count the number of iterations), and the experimental evaluation: runtime analysis is a very important measure of the utility of an algorithm, and this is missing for the real-world datasets.

More details below (in no particular order):

1. The computational complexity does not consider the number of iterations before convergence.  Therefore I find the claim that the algorithm takes $O(nc)$ time (or rather $O(n(k + c))$ where $c$ is the number of clusters and $k$ is the number of distances known per point) misleading or incorrect.  In fact, Lloyd's algorithm also takes $O(nc)$ per iteration, but the number of iterations is not easily bounded.  Unless the authors can conclusively show that the number of iterations should not be considered in the runtime, this claim must be modified or retracted.

2. As pointed out in the introduction, the initialization of k-means can make a huge difference for performance.  However, on line 204 the authors simply write "K-means is initialized in a random way"---but what is this random way?  If I were to, e.g., randomly assign points to clusters to initialize, this can lead to very poor results and slow convergence, at least for Lloyd's algorithm.  Even
running many trials of this strategy will produce poor performance for every strategy.  Thus proper initialization is very important.  It would be very helpful if the authors could provide more information on this point.

3. The authors have not provided any information on runtime on real-world datasets.  Thus it is difficult to evaluate the utility of the proposed algorithm, LKM.  Clustering algorithms often behave very differently on diverse datasets, and thus I think that runtimes only on the toy dataset---and only comparing with Lloyd's algorithm---does not give a full picture.  In fact, the number of iterations on the toy datasets is very small when compared to what Lloyd's algorithm will do on higher-dimensional real datasets (where cluster boundaries are not so clear-cut).  For the sake of clarity, the authors should address the issue of runtime comprehensively on real-world datasets.

4. It would also be nice to include some information on the dimensionality and number of points in each dataset listed in 4.2.1 in the main paper, if possible.  I appreciate that 16 datasets are used---this seems like a fairly comprehensive set of experiments---but I suspect most readers will not have an idea of what the characteristics of these datasets are, and many will not look at the
supplemental material.  When I consult the supplementary material, I see that many of the datasets used are relatively high-dimensional; the minimum dimension appears to be 256.  I would suggest that the authors also try on lower-dimensional data (e.g. below 50 dimensions), as convergence characteristics of different algorithms tend to be very different in lower dimensions.

5. A note about the limitations: there are many techniques to quickly construct k-NN graphs.  In lower dimensions, this can be done via trees (see, e.g., [1-3]); in higher dimensions, this can be done approximately via hashing (e.g., [4-5]).  In fact even if only a similarity function is used, so long as that function is a kernel, additional branch-and-bound techniques could be used ([6-8]).  Anyway, perhaps you will find these interesting, if you were not already aware of those lines of work.

6. When comparing Lloyd's algorithm to LKM, note also that there are several variants of Lloyd's algorithm that produce exactly the same result in accelerated time.  The authors even cite some (reference [11]).  There are more however, especially Elkan's [9] and Hamerly's [10] that perform very well in practice that should probably be considered in an empirical comparison.  Personally, if I am doing k-means in practice, I often don't bother with the standard Lloyd algorithm because Elkan's or Hamerly's variants (or the
tree-based ones) are so much faster---it can often be an orders-of-magnitude speedup.

7. Can more detail be provided in Algorithm 1 on how y is initialized randomly?  Is the algorithm uniformly randomly drawing a cluster assignment for each point?

8. I think the paper could benefit from some careful proofreading.  There are numerous grammatical issues, especially having to do with tenses (present/past/etc.).

9. The suggestion in Eq. (10) to convert a (non-metric) similarity function to a dissimilarity function still produces a non-metric: it is not guaranteed to satisfy the triangle inequality.  This makes the change from Eq. (8) (which is built on distances using D) to Eq. (11) (which is built on dissimilarities using G) significant: particularly, I am no longer certain that we are solving the same problem as (3).  I am okay with the fact that LKM solves a relaxation of (3) but I think this should be explicitly pointed out.  On the other hand, note that if the similarity function is a kernel, then it *does* induce a distance metric in kernel space, and that could be used instead.  Specifically, Eq. (2.2) in [8] uses this induced distance metric.

[1] "N-Body'problems in statistical learning", NIPS 2001
[2] "Tree-independent dual-tree algorithms", ICML 2013
[3] "Random projection trees and low dimensional manifolds", STOC 2008
[4] "Near-optimal hashing algorithms for approximate nearest neighbor in high
dimensions", FOCS '06
[5] "LSH Forest: Practical Algorithms Made Theoretical", SODA 2017
[6] "Asymmetric LSH (ALSH) for sublinear time maximum inner product search (MIPS)", NIPS 2014
[7] "Maximum inner-product search using cone trees", KDD 2012
[8] "Fast Exact Max-Kernel Search", SDM 2013
[9] "Using the triangle inequality to accelerate k-means", ICML 2003
[10] "Making k-means even faster", SDM 2010

I hope that these comments are helpful!

**Time Spent Reviewing:**

2

---

> ### Author Response · Authors · 2021-08-09
> **The computational complexity and the way of initialization.**
>
> ## For problem 1:
> For each algorithm, the computational complexity per iteration is analyzed in this article.
> LKM takes $O(t_1 n k)$ time and Lloyd’s algorithm takes $O(t_2 n d c)$ time if take the number of iteration into account, where $t_1$ and $t_2$ denote the number of iterations of LKM and Lloyd's, $n$ is the number of samples, $c$ is the number of clusters, and $k$ is the number of distances known per point.
> Generally speaking, $k$ is much smaller than the product of $d$ and $c$. In addition, it can be seen from Table 1 that our algorithm (LKM) usually converges faster than Lloyd’s.
>
> ## For Problem 2:
>
> ### What is the random way?
> Let vector y denote the clustering result, y_i is a random integer from the “discrete uniform” distribution in the interval [1, c]. For more detailed information see "numpy.random.randint" (a function in "numpy") please.
>
> ### A proper initialization is very important for K-means, but why the random way is used?
> - Although there are many methods proposed for the initialization of k-means, these methods will not necessarily improve the performance of other baselines.
> - Most baselines involve initialization. In order to ensure fairness in the experiment, all algorithms should use the same initialization.
> - Our algorithm achieves a promising performance with the random initialization, which verifies the robustness of LKM to initialization.
>
> ## For problem 3:
> See the reply to reviewer 8H3D (For the problem mentioned in paragraph 1).
>
> ## For problem 6:
> From the reply to reviewer 8H3D, we can see that $k$-means only serves as an intermediate medium to verify the efficiency of LKM, so it is not necessary to consider those fast k-means algorithms.
>
> I will seriously consider the valuable suggestions in 4, 5, 7, 8, and 9, and carefully modify the submission, regardless of whether the submission is accepted or not.
>
> Let me know if you have any other questions.

---

> > ### Comment · Reviewer_bmZT · 2021-09-10
> > **Reply**
> >
> > Unfortunately I cannot improve my score based on the discussions that have happened here.
> >
> > I don't agree with the long discussion in the response to reviewer 8H3D to mostly ignore large real-world datasets; and I also disagree with the claims to simply ignore accelerated k-means variants like Elkan's and Hamerly's algorithms. Pretty much any clustering algorithm is a tradeoff between clustering efficacy and runtime. LKM seems to sit somewhere that is more efficient than graph-based clustering algorithms but less effective, and more effective than k-means but slower. However, the paper does not demonstrate (at least to my satisfaction) that there is a regime in which the tradeoffs of LKM actually make sense compared to the existing state of the art---and that requires consideration of accelerated k-means variants to give a full comparison, especially since that plus choice of initialization strategy could make the k-means runtime results look completely different in Table 1 in the response to reviewer 8H3D.
> >
> > I would encourage resubmission, once the claims and arguments are cleaned up somewhat to clarify all the various questions that have come about from the reviews.

---

> > > ### Author Response · Authors · 2021-09-14
> > > **Time(LKM) < Time(k-means) < Time(graph-based methods)**
> > >
> > > First of all, I respect your rating of this article.
> > >
> > > The main controversy between us lies in the efficiency of LKM. To this end, I first declare the following five points:
> > >
> > > 1. The proposed algorithm LKM is a **graph-based** clustering algorithm.
> > > 2. From Table 3, we can see that LKM has achieved the **highest performance** comparing to several graph-based clustering algorithms in most cases, in terms of **ACC, NMI, and ARI**.
> > > 3. From Table 2 and Table 4, we can see that LKM achieved the **shortest running time** comparing to $k$-means.
> > > Table 4 means Table 1 that shown in the response to reviewer 8H3D
> > > 4. The time complexity of $k$-means is **much lower** than that of those graph-based clustering methods included in baselines.
> > > 5. Based on the discussion in 3 and 4, we conclude that LKM has a time advantage over these graph-based algorithms.
> > >
> > > The dispute between us mainly lies in the Point 4 and 5, if I am not wrong.
> > >
> > > To verify our proposition, we report the running times of these baselines on real-world datasets. From the results in Table 5, we can see that $k$-means is more efficient than these graph-based algorithms, and LKM is more efficient than $k$-means, as mentioned earlier.
> > >
> > > In addition, I can’t agree with your following sentence
> > > “LKM seems to sit somewhere that is more efficient than graph-based clustering algorithms but less effective, and more effective than $k$-means but slower.”
> > > The meaning of this sentence is **completely inconsistent** with the experimental results, I think there must be a misunderstanding here.
> > >
> > > So far, **we have verified the efficiency and effectiveness of LKM comparing to several graph-based methods**. Therefore, we believe that it is not necessary to compare LKM with accelerated $k$-means.
> > >
> > >
> > > Table 5  The running time(s) on real-world datasets. (The time taken to construct the k-NN graph is also **counted**).
> > >
> > > |                |   AGCI   |   FCDMF  |  FINCH | k-means | KSUMS |   RCC   |    SC    |  LKM  |
> > > |:--------------:|:--------:|:--------:|:------:|:-------:|:-----:|:-------:|:--------:|:-----:|
> > > |   LFW (13233)  | 7185.216 | 15251.89 | 25.665 |  65.02  |  9.73 | 415.707 | 3301.298 | 9.931 |
> > > |  CALFW (12174) | 3281.551 | 6860.643 | 24.115 |  60.647 |  9.1  | 209.309 | 1580.974 | 8.681 |
> > > |  CPLFW (11652) | 2879.552 | 6106.754 | 29.112 |  43.63  |  6.46 | 217.154 | 3822.206 | 6.742 |
> > > |  FaceV5 (2500) |  18.548  |  23.167  |  2.576 |   1.52  | 0.529 | 130.113 |   5.776  | 0.409 |
> > > |   CFPW (7000)  |  60.244  |  73.396  | 13.919 |  8.788  | 3.296 | 111.112 |  32.611  | 2.828 |
> > > |  CMUPIE (2856) |  14.149  |   10.42  |  8.756 |  3.505  | 0.724 | 392.289 |   1.062  | 0.538 |
> > > |   Colon (62)   |   0.097  |   0.016  |  0.088 |  0.057  | 0.001 |  0.422  |   0.044  | 0.002 |
> > > |  Dexter (600)  |   0.641  |   0.559  |  0.301 |  0.118  | 0.026 |  8.722  |   0.078  | 0.029 |
> > > |  FERET (11338) |  185.106 |  203.777 | 24.298 |  20.358 | 7.681 | 371.136 |  153.738 | 7.362 |
> > > |   GTdb (750)   |   0.741  |   0.656  |  0.611 |  0.137  | 0.086 |  15.848 |   0.118  | 0.065 |
> > > | Madelon (2600) |   8.442  |   7.683  |  2.486 |  0.476  | 0.626 | 152.158 |   0.643  | 0.416 |
> > > |  Mpeg7 (1400)  |   4.615  |   3.697  |  1.181 |  0.601  | 0.198 |  44.162 |   0.265  | 0.149 |
> > > |   MUCT (3755)  |  20.041  |   21.27  |  4.826 |  1.535  | 0.843 |  58.203 |   3.181  | 0.778 |
> > > | Olivetti (400) |   0.185  |   0.148  |  0.202 |  0.063  | 0.032 |  1.906  |   0.067  | 0.024 |
> > > |   Umist (575)  |   0.568  |   0.458  |  0.51  |   0.3   | 0.065 |  21.224 |   0.073  | 0.031 |
> > > |   Yale (165)   |   0.218  |   0.098  |  0.167 |  0.148  | 0.008 |  3.751  |   0.04   | 0.007 |
> > >
> > > The number in parentheses indicates the number of samples in the dataset.

---

### Official Review · Reviewer_PD9K · 2021-07-13

**Rating:** 7
**Confidence:** 4

**Summary:**

This submission introduces a computationally efficient clustering algorithm called LKM, which focuses on the scenarios where data is presented in the form of graph. In brief, only the k-nearest neighbour distances of each point are considered in the objective and a simple truncation of large distances is introduced to mitigate the effect of outliers.  The method is shown to produce promising practical performance in comparison with existing techniques both computationally and in terms of clustering performance.



**Ethical Concerns:**

N.A.

**Ethics Review Area:**

["I don’t know"]

**Limitations And Societal Impact:**

Yes.

**Main Review:**

Strengths:

LKM is developed based on the k-means problem which is a fundamental problem in machine learning and the authors might bring some new insights and develop a new algorithm. The main strength of LKM lies in the practical aspects of the proposed algorithm, in that it is efficient and the performance is compelling.
Specifically, the computational overhead is O(nk), where n and k denote the number of samples and nearest neighbors, respectively.

In addition, the unified framework of KSUMSC, and LCM also seems interesting. It revolves that SC, a popular clustering algorithm derived from graph cuts, can be optimized with the same time complexity as that of LKM.

Weaknesses:
1. Line 9: "Specifically....are denote the...". The sentence is grammatically incorrect, "are" should be removed.

2. Some works relating to the connection between k-means and clustering with graph cuts are not discussed in Section 2 (Related work).

3. The performance of SC optimized by the proposed algorithm is not shown.

4. What does \bar{y} stand for in Eq. (15) and (16)?

5. e_i should be a column vector.

6. Some steps in the derivations are not explained. What is the connection between (11) and (17)?

7. The evaluation needs to be improved. The proposed clustering method uses ball-tree to construct a k-nearest neighbors graph as an intermediate step. Correspondingly, it is suggested to include the clustering performance of ball-tree as a baseline to compare.

8. Have the paper proof-read to remove grammatical errors

9. In page 7, line 178, "#Iter" should be "#Iter.".

10. In page 7, table 2, "FLK" should be "LKM".

11. References should be shown in the same format.

**comment after rebuttal**
I acknowledge that I read and other reviewers' comments and authors' responses. I would like to keep my score as most concerns of mine have been well addressed.



**Time Spent Reviewing:**

3

---

> ### Author Response · Authors · 2021-08-09
> **Thanks**
>
> Thank you for your affirmation and encouragement. Your suggestions are of great help to the improvement of the paper. I will seriously consider these suggestions, and carefully modify the submission, regardless of whether the submission is accepted or not.

---

### Official Review · Reviewer_o7mN · 2021-07-15

**Rating:** 5
**Confidence:** 4

**Summary:**

This paper considers a variant of k-means which takes the k-nearest neighbor graph as input. The proposed algorithm, called local k-means (LKM in short), then works by locally optimizing the assignment vector of a point using an objective function that is roughly the incremental cost for the k-means objective. The paper provided a lot of empirical data to show the advantages of LKM.

**Limitations And Societal Impact:**

The authors have adequately addressed the limitations and potential negative societal impact of their work.

**Main Review:**

Overall, this paper proposes a novel k-means algorithm, but I think there has not been enough discussion on its guarantees (from both a theoretical and empircal point of view) to make it convincing. The paper writing can also be greatly improved.

Originality: The mthods used in this paper are new. The paper converts the k-means objective into a different format which is compatible with nearest-neighbor graph inputs, where the main variable is the indicator matrix, and optimizes the columns in this matrix one by one (each column corresponding to the assignment vector of one point). The idea of optimizing the assignment of each point to clusters might not be new to the community, but the proposed local optization function seems to be novel.

Quality: The paper does not provide any theoretical guarantees for the proposed algorithm and mainly used empirical evaluation approaches to show its superiority. The experiments are carefully designed, but I think the authors are trying to make some claims that are too broad and thus the provided empirical results do not seem to be convincing enough to show that this approach really beats state-of-the-art. LKM is compared with a lot of baselines and that may still not cover the entire state-of-the-art clustering approaches. Some comments:
- The self-synthesized data are well designed, and serves to show that LKM performs well on some datasets whose natural clustering cannot be captured by the optimal k-means objective function. However, it seems to me that they might be suitable for other types of clustering such as density-based clustering (DBSCAN might be good for "Mickey"), or common k-means implementation packages might also be good for them (k-means++ might be good for "Outlier" depending on the sampled seeds?).  I wonder if the authors have considered these?
- After reading the experiment section I get a bit more confused about the claims that this paper is trying to make: LKM seems to be beating most baselines on the datasets in clustering quality and be significantly faster than k-means? The improvement of LKM , especially when compared with KSUMS on most of the datasets, do not seem to be very significant. Actually, the results gave me the impression that there are many datasets and each of them might be suitable for different types of clustering methods, with LKM performing relative well for all datasets.

Clarity: The writing clarity is fine, although occasionally there are some flaws such as definition/notation that has not been used again later, typos and so on, but it does not affect reading too much. It has a clear problem definition and algorithm description. That being said, I'm not very impressed with the organization of the paper.
- The paper spent enough space on problem definition but the paragraphs did not serve to fully motivate this design. For example, previous methods like KSUMS and LKM, whose math formulation, as the authors pointed out, have very similar structure with the one used in this paper, but the authors did not provide intuition about why the differences lead to better performance.
- The description of the algorithm design is only about one page and I think there should be more intuitive explanation of why this particular design of local objective function (18) might lead to a better solution and make it robust against outliers.

Significance: This paper could potentially propose a solution that advances state-of-the-art, but the comparsion between the new algorithm and previous approaches does not seem to convince people of its superiority. Also, I'm not sure if the improvement is significant enough for this venue.

Minor comments:
- In abstract, line 5, the authors mentioned "KSUMS and SC" without citation and before defining which algorithms these are referring to.
- line 26, "For matrix A, we call it indicator matrix", I don't really see the point of introducing the indicator matrix here, and the notation is not really used again in this paper.
- line 86, "where diag(A) = [a11,..., ann]^T",  A is not used in (7), and I'm not sure about what a11, ..., ann are referring to.
- In (9), I'm not sure about why is d_ij^{(k)} set as \gamma otherwise? Can the authors comment on this?


**Time Spent Reviewing:**

5 hrs

---

> ### Author Response · Authors · 2021-08-09
> **In this response, we focus on the following two points: 1. Why not consider the density-based methods. 2. The purpose of the experiments**
>
> ## For the first problem:
> ### Why not consider the density-based methods?
> The scenario that density-based algorithms focus on is different from those algorithms discussed in this paper. Specifically, density-based algorithms should be used if we don't know the number of clusters, but the number of clusters in the scenario we are concerned about is known in advance and is fixed. If the DBSCAN algorithm is used on the "Mickey", it is very likely that the number of clusters generated may be larger than or less than 3, which is what we do not want to see.
>
> ### k-means++ might be good for "Outlier" depending on the sampled seeds?
> No matter what initialization is used, the performance of $k$-means is poor on "Outlier". Therefore, $k$-means++ will fail on this dataset.
>
> Suppose that $k$-means is initialized with ground truth labels，as shown in Figure 3(a). Then we have the following four centers: orange: (0, 0), green: (0, 5), blue: (5, 0) and purple: (10, 10).
> Without point A (100, 100), the center of the purple cluster with size 20 is (5, 5). Thus, the coordinate of the purple cluster with A can be calculated by
> \begin{equation}
> x=(5\*20 + 100) / 21 \\approx 10, \\\\
> y=(5\*20 + 100) / 21 \\approx 10.
> \end{equation}
> It is not difficult to find that the samples (exclude  A) belonging to the purple cluster will be assigned into the blue or green cluster, which is the reason why $k$-means failed.
>
> ## For the second problem:
> This article proposed a graph-based algorithm named LKM. It can be optimized efficiently and its performance is comparable with several state-of-art graph-based clustering methods.
>
> ### Efficiency
> The experiment conducted on large-scale synthetic datasets is used to verify the efficiency of LKM, A more detailed explanation can be found in the reply to reviewer 8H3D.
>
> ### Effectiveness
> From the results shown in Table 3, we can see the performance (ACC, ARI, etc.) of LKM is comparable with several clustering algorithms, which does not seem to be controversial.
>
> ## For Minor comments:
> - Indicator matrix is a concept often involved in clustering. For example, the matrix $Y$ defined in Eq. (2) is an indicator matrix.
> - $diag(A)$ is seen as a function here. Its input in the formula is $Y^TDY$ or $(Y^TY)^{-1}$.
> - If $x_i$ does not belong to $N_k(x_j)$, then the distance between $x_i$ and $x_j$ is usually larger. So we use a larger value $\gamma$ to replace these distances. It is the maximum value of set $\\{\Vert x_i -x_j \Vert _2^2 \mid x_i \in N_k(x_j), i=1, \cdots, n \\}$.

---

### Official Review · Reviewer_8H3D · 2021-07-17

**Rating:** 6
**Confidence:** 5

**Summary:**

This paper introduces a local efficient version of k-means which leverages the KNN graph of the data to compute neighborhoods of data. Extensive empirical results both on synthetic and real data are reported.



**Limitations And Societal Impact:**

Limitations related to the construction of the KNN graph are discussed. No potential negative societal impact is expected.

**Main Review:**

Running times are not reported for the results on real data in Table 3. One of the main advantages of the proposed method is gained efficiency with respect to the clustering competitors, therefore running times should be reported and discussed.

The proposed method LKM also depends on the initialization of the clustering y, and this is not discussed. It's unclear whether the results on simulated data in Table 1 are averages of different runs or not. This is not discussed and it should. Results should be averages and standard deviations should also be reported.

It is well-known that k-means is sensitive to outliers, but it's unclear in Figure 3(b) why the performance of k-means is poor in this case. It seems it was still able to detect the four clusters in the data?

In Section 3, the definition of neighborhood of a point is confusing and not necessarily correct. The notion of neighbor is not symmetric: x_j may be a neighbor of x_i, but not vice versa.

I don't find the analogy of KNN graph with social networks particularly useful or insightful. They are not quite the same concept, really. The paper can instead be improved by discussing related work on exemplar-based approaches aimed at making clustering more robust. See for example recent work on regularized NMF methods which leverage the KNN graphs of the data, e.g. Mani et al, Unsupervised Selective Manifold Regularized Matrix Factorization, SDM 2021, and related work.


**Time Spent Reviewing:**

2 hours

---

> ### Author Response · Authors · 2021-08-09
> **In this reply,  we focus on the following two things: 1. What do we do to verify the efficiency of LKM? 2. Why $k$-means is failed on "Outlier"?**
>
> ## For the problem mentioned in paragraph 1.
>
> This article proposed a graph-based algorithm named LKM. It can be optimized efficiently and its performance is comparable with several state-of-art graph-based clustering methods. However, the absence of running time in Table 3 makes the efficiency of LKM controversial.
>
> ### Why are the running times not shown in Table 3?
>
> It is meaningless to compare the running time on these datasets, as the number of samples of them is not very large and the running time is not dominated by the number of samples.
>
> What do we do to verify the efficiency of LKM?
> It is difficult to directly compare the running time of LKM and other graph-based algorithms on large-scale datasets, which will be explained later. However, most graph-based algorithms have a higher computational complexity than $k$-means [1] (Section 5.3) and [2] (Section II. A). Thus, we compare LKM and $k$-means on some large-scale synthetic datasets. From the results shown in Table 2, we can see that LKM has a significant improvement in terms of the running time, compared to $k$-means. It can be used to indirectly illustrate the efficiency of LKM.
>
> ### Why not compare LKM and baselines on large-scale benchmark datasets?
> - If LKM significantly outperforms other graph-based algorithms in terms of running time but significantly worse in terms of accuracy, then this result is meaningless and cannot verify the efficiency of LKM. So, when verifying the superiority in terms of running time, the accuracy should also be taken into account.
> - Since LKM is a graph-based clustering algorithm, comparison with some graph-based algorithms is needed to verify the effectiveness.
> - Most graph-based algorithms are failed to get a result since the running time is too long or the required memory is too large. So, we cannot verify the effectiveness of LKM since the absence of the performance of baselines. Thus, the experiment result is not persuasive, as discussed above.
> - In Table 2, only $k$-means is taken as the baseline, why the results of this experiment are convincing? In Table 2, the datasets constructed are suitable for $k$-means [3]. The effectiveness of LKM can be verified if it achieves performance better than $k$-means's on these datasets.
>
> In addition, we conducted experiments on two large-scale datasets.
>
> Table 1: Running time (s) of LKM and $k$-means.
>
> |     Dataset          |     # Samples    |     #Features    |     # Subjects    |     EFANNA    | KSUMS |     $k$-means    |     LKM    |
> |----------------------|------------------|------------------|-------------------|---------------|-------|------------------|------------|
> |     CASIA_WebFace    |     494414       |     256          |     10575         |     93.70     | 110.4 |     953.9        |     **44.3**   |
> |     CelebA           |     202599       |     256          |     10177         |     25.00     | 31.2  |     155.0        |     **13.4**   |
>
> EFANNA[5] is the algorithm that is used to generate the $k$-nearest neighbors graph.
>
> Table 2: Performance of LKM and $k$-means. (PRE means precision, REC means recall, F1 means $F_1$ score, and KM means $k$-means)
>
> |     Dataset    |     PRE(KSUMS)    |     PRE(KM)    |     PRE(LKM)    |     REC(KSUMS)    |     REC(KM)    |     REC(LKM)    |     F1(KSUMS)    |     F1(KM)    |     F1(LKM)    |
> |----------------|-------------------------|----------------------|-----------------------|----------------------|-------------------|--------------------|------------------|---------------|----------------|
> |     WebFace    |     0.644               |     0.546            |     **0.730**             |     0.572            |     0.547         |     **0.694**          |     0.606        |     0.546     |     **0.712**      |
> |     CelebA     |     0.445               |     0.404            |     **0.450**             |     0.486            |     **0.560**         |     0.520          |     0.464        |     0.468     |     **0.483**      |
>
> As we have seen, in terms of efficiency, LKM outperforms $k$-means on large-scale data.
> On large-scale datasets, most graph-based algorithms cannot produce results because of their high computational complexity.
>
> **There are also some explanations about this issue in the reply to Reviewer bmZT**.
>
> ---
> ## For the problem mentioned in paragraph 2.
>
> All algorithms involved in Table 1 were run 50 times, and the average results were reported, as mentioned in line 182, page 7.
> The standard deviations of LKM are shown in the supplementary material (Section 4).
> The standard deviations on synthetic datasets are not reported, but I think it is similar to that on benchmark datasets and will show it in the subsequent version.
> ---
> ## For the problem mentioned in paragraph 3.
>
> Suppose that $k$-means is initialized with ground truth labels，as shown in Figure 3(a). Then we have the following four centers: orange: (0, 0), green: (0, 5), blue: (5, 0) and purple: (10, 10).
> Without point A (100, 100), the center of the purple cluster with size 20 is (5, 5). Thus, the coordinate of the purple cluster with A can be calculated by
> \begin{equation}
> x=(5\*20 + 100) / 21 \\approx 10, \\\\
> y=(5\*20 + 100) / 21 \\approx 10.
> \end{equation}
> It is not difficult to find that the samples (exclude  A) belonging to the purple cluster will be assigned into the blue or green cluster, which is the reason why $k$-means failed.
> ---
> ## For the problem mentioned in paragraph 4.
>
> In this paper, we define $N_k(x_i)$ = {$x_j$ | $x_j$ is among the k-nearest neighbors of $x_i$ or $x_i$ is among the k-nearest neighbors of $x_j$} as neighbors of $x_i$, so, the problem you mentioned will not appear in this article.
>
> I will seriously consider the suggestions and carefully modify the submission, regardless of whether the submission is accepted or not.
>
> Let me know if you have any other questions.
>
> [1] X. Chen, J. Z. Haung, F. Nie, R. Chen and Q. Wu, "A Self-Balanced Min-Cut Algorithm for Image Clustering," 2017 IEEE International Conference on Computer Vision (ICCV), 2017, pp. 2080-2088, doi: 10.1109/ICCV.2017.227.
>
> [2] D. Cai and X. Chen, "Large Scale Spectral Clustering Via Landmark-Based Sparse Representation," 2015 in IEEE Transactions on Cybernetics, vol. 45, no. 8, pp. 1669-1680, Aug. 2015, doi: 10.1109/TCYB.2014.2358564.
>
> [3] F. Nie, X. Wang, and H. Huang, "Clustering and Projected Clustering with Adaptive Neighbors," 2014 In Proceedings of the 20th ACM SIGKDD international conference on Knowledge discovery and data mining (KDD), 2014, pp. 977–986, doi: 10.1145/2623330.2623726
>
> [4] C. Fu and D. Cai, "Efanna: An extremely fast approximate nearest neighbor search algorithm based on knn graph," 2016 arXiv preprint arXiv:1609.07228

---

### Decision · Program_Chairs · 2021-09-27

**Decision:**

Reject

**Comment:**

This paper introduces faster algorithms for clustering.  The paper has some good insights, but ultimately the reviewers were not convined that there is a regime in which the tradeoffs of LKM (their algorithm) actually make sense compared to the existing state of the art.  Due to this, the paper falls below the competition at NeurIPS.